# Saccade suppression depends on context

Eckart Zimmermann*

Institute for Experimental Psychology, Heinrich Heine University Düsseldorf, Düsseldorf, Germany

**Abstract** Although our eyes are in constant movement, we remain unaware of the high-speed stimulation produced by the retinal displacement. Vision is drastically reduced at the time of saccades. Here, I investigated whether the reduction of the unwanted disturbance could be established through a saccade-contingent habituation to intra-saccadic displacements. In more than 100 context trials, participants were exposed either to an intra-saccadic or to a post-saccadic disturbance or to no disturbance at all. After induction of a specific context, I measured peri-saccadic suppression. Displacement discrimination thresholds of observers were high after participants were exposed to an intra-saccadic disturbance. However, after exposure to a post-saccadic disturbance or a context without any intra-saccadic stimulation, displacement discrimination improved such that observers were able to see shifts as during fixation. Saccade-contingent habituation might explain why we do not perceive trans-saccadic retinal stimulation during saccades.

## Introduction

The sensorimotor contingency theory proclaims that perception consists in mastering the systematic relations between actions and their perceptual consequences (*O'Regan and Noë, 2001*). The prime example for sensorimotor contingencies are those movements we perform most often in real life, that is saccade eye movements and the associated motion stimulation they produce. Every time we perform a saccade the external world sweeps with high speed motion across the retina. Consequently, the visual system has to deal with two challenges: First, to keep in register where in external space each pre- and post-saccadic image originated from, and second, to suppress any motion sensation that would arise during the retinal displacement.

Learning sensorimotor contingencies requires a signal that indicates the initiation of a movement. In eye movement research, the concept of the efference copy arose, that is a signal which informs visual areas about the metrics of upcoming eye movements (*Wurtz, 2018*). To guarantee the spatial alignment across saccades between internal and external space, neurons in the intra-parietal cortex (*Duhamel et al., 1992*), in the frontal eye fields (*Umeno and Goldberg, 1997*), the superior colliculus (*Walker et al., 1995*) and in visual areas V2 and V3 (*Nakamura and Colby, 2002*) shift their receptive fields predictively, triggered by an efference copy signal (*Sommer and Wurtz, 2006*). The suppression of intra-saccadic motion perception might be driven by such an extra-retinal signal likewise (for a review, see *Binda and Morrone, 2018*).

As it has been shown that the visual system is able to detect high-speed motion (*Burr et al., 1982*), the question arises why we are not aware of the motion stimulation produced by a saccade. In the search for a mechanism that cancels trans-saccadic motion perception, researchers have found that sensitivity of visual contrast (*Volkmann et al., 1978*) and of motion (*Burr et al., 1982*; *Shioiri and Cavanagh, 1989*; *Ilg and Hoffmann, 1993*) drops down at the time of saccades. This process - called saccade omission - amounts to 0.5–1 log units, is strongest for low spatial frequencies (*Volkmann, 1986*; *Burr et al., 1994*) and is homogeneous across the visual field (*Knöll et al., 2011*). It starts ~50 ms before saccade initiation, peaks around saccade onset and fades away 50 ms later (*Volkmann et al., 1978*). It has been argued that the intra-saccadic decrease of visual sensitivity

**\*For correspondence:**
eckart.zimmermann@hhu.de

**Competing interests:** The author declares that no competing interests exist.

is the signature of a mechanism that shuts down the magnocellular pathway at an early neural level to prevent the disturbing motion experience during saccades (*Burr et al., 1994*).

However, in contrast to an early shut-down resulting in motion blindness, *Castet and Masson (2000)* have demonstrated that motion detection during saccades is still possible if stimuli are optimized for high-speed motion detection. They used low-contrast gratings drifting at a constant high speed (360 dva/s) and found that the ability to perceive intra-saccadic motion depended on the difference between the peak eye velocity and grating speed. In their view, the magnocellular pathway is still active at the time of saccades and the peri-saccadic reduction of contrast sensitivity results from retinal factors (*Castet et al., 2002*). In a follow-up study the same authors could further show that this ability to perceive fast intra-saccadic motion declined when additional pre - and post-saccadic stimuli were visible. They suggested that these static images masked intra-saccadic motion thus reducing awareness.

An earlier report already showed that masking might cancel the retinal smear produced by the saccade (*Campbell and Wurtz, 1978*). *Duyck et al. (2016)* recently devised a method to test the perception of intra-saccadic smear objectively and thereby corroborated the idea that masking hides saccade-induced motion from perception. In that study smear-masking survived a dichoptic presentation thus suggesting a central origin. If masking and not saccade-related mechanisms explains saccadic omission, one should find masking-induced suppression even in the absence of saccades. Recently, *Duyck et al. (2018)* used simulated saccades and found motion perception to be much less salient in the presence of static objects.

While masking provides a very parsimonious account of how the retinal smear remains hidden from awareness, it is still an open question why we do not consciously perceive a motion transient from the pre-saccadic to the post saccadic image. It is long known that trans-saccadic displacements of a certain size go unnoticed by the observer (*Bridgeman et al., 1975*; *Niemeier et al., 2003*), a phenomenon termed saccadic suppression of displacement. The reason for the poor displacement detection is that the saccade covers the motion transient because the presence of motion transients leads to very high displacement sensitivity (*Legge and Campbell, 1981*).

Here, I offer a novel explanation of suppression which states that the brain stores and habituates to sensorimotor contingencies. This account does not require elimination of peri-saccadic perception via contrast reduction at an early processing level. I show that the saccadic suppression results from a habituation to intra-saccadic stimulation (see *Figure 1*). In this view, neurons in an extra-retinal storage mechanism will be informed by an efference copy about the initiation of a saccade, store saccade-induced visual stimulation and saturate their response to the visual information most dominant in the previous history of the last set of saccades that has been executed.

This idea predicts that suppression magnitude can be modulated by the visual context in which saccades are performed. Performing saccades in a structured environment induces recurring intra-saccadic stimulation. In order to create an experimental environment in which intra-saccadic information could be systematically varied, I used a horizontal grating as a background on which horizontal saccades had to be performed. As the grating was oriented parallel to the saccade path it should have produced only negligible retinal motion information. Context-sensitivity would predict that performing many saccades on that background should lead to a minimal magnitude of saccade suppression compared to natural vision, where saccades are performed mostly across structured environments.

The intra-saccadic stimulation was displayed as soon as a saccade was detected. I used a grating displacement orthogonal to the rightward saccades, that is upward displacements of the horizontal grating, in order to disentangle the artificially produced shift (the grating displacement) from the retinal motion. Testing displacement discrimination parallel to the saccade path requires more sophisticated experimental designs (*Castet and Masson, 2000*) that would have led to an unjustifiable number of trials. If the sensorimotor system takes into account this exposure, intra-saccadic displacement detection should drastically decline. Consistent with context-sensitivity, saccade suppression magnitude was strong when observers performed many trials in which stimulation was presented contingent on saccade execution. However, saccade suppression was absent if the background grating remained stationary and observers were able to perceive intra-saccadic motion as well as during fixation.

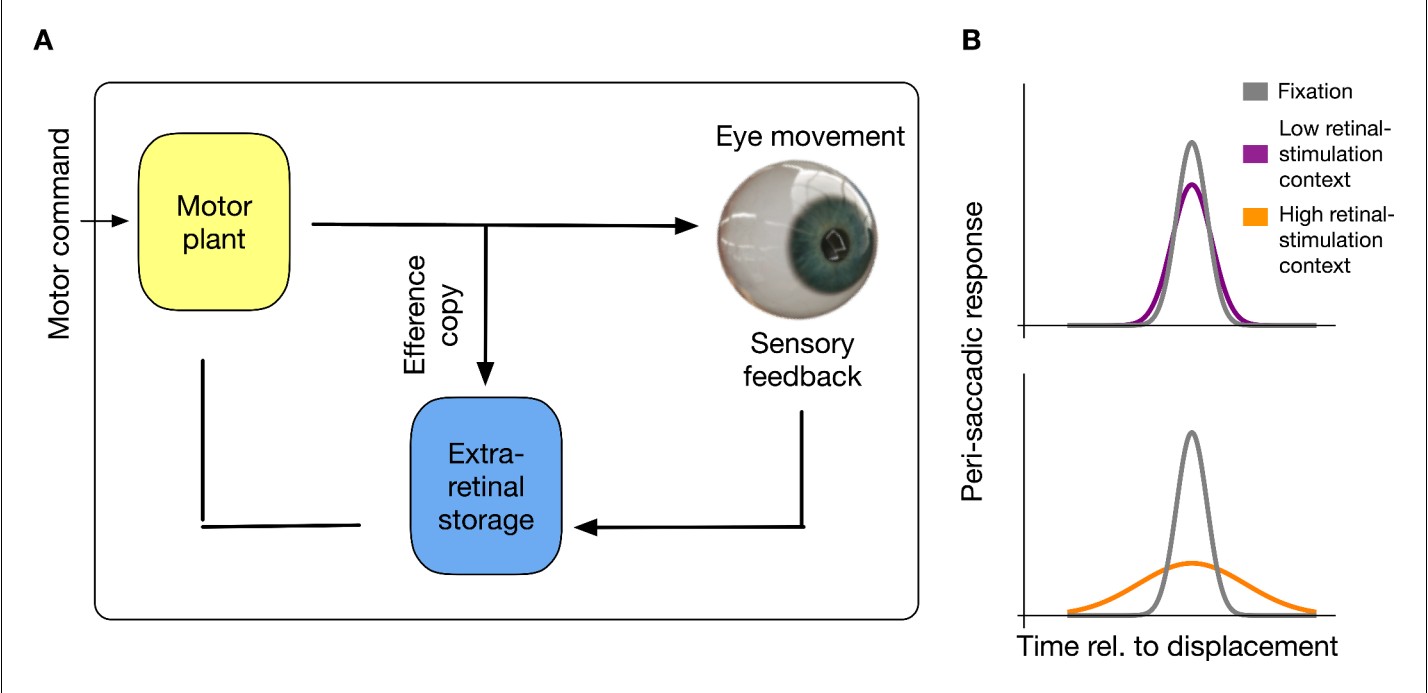

**Figure 1.** Intra-saccadic habituation to sensorimotor contingencies. (**A**) Schematic illustration of the proposed model. For saccade generation, a motor command is sent to the motor plant which produces the desired eye movement. Intra-saccadic visual stimulation will activate an extra-retinal storage mechanism. Importantly, this storage mechanism is activated only under the combined condition that an efferent copy signals the occurrence of an eye movement and visual stimulation is sensed. (**B**) Recurring input of visual stimulation will habituate neurons in the extra-retinal storage, consequently leading to a reduced sensitivity for intra-saccadic stimulation. In a context with low retinal stimulation, habituation remains weak and intra-saccadic neuronal responses to visual stimulation remain comparable to fixation. However, in a context with high retinal stimulation neurons habituate and intra-saccadic neuronal responses become weak.

## Results

I asked participants to perform horizontal, rightward saccades in a dark environment. Stimuli were presented on a computer screen that was covered by a semi-transparent foil in order to dampen visual stimulation as much as possible. The background consisted of a sinusoidal horizontal grating shown with a spatial frequency of 0.05 c/dva (see *Figure 2A*). At the start of each experimental session (except baseline), participants performed 105 saccades over the grating. In these context trials - depending on the session - the grating was displaced either during or after the saccade or it remained stationary.

Shifting the grating during saccade execution allowed to systematically manipulate intra-saccadic stimulation. *Figure 2B* shows the time-course of events for sessions in which the grating was shifted upward as soon as the eye-tracker detected saccade initiation. After a fixation period of 1000–1500 ms the saccade target was presented only briefly and participants started their saccades after the disappearance of the fixation point. Since the experiment was conducted in a dark room and the monitor screen was covered by a semitransparent foil, no visual information except the shifting grating was present during saccade execution. I checked the timing of the grating displacement relative to saccade performance in an offline analysis. Depending on sessions, displacements were presented well within (see *Figure 2C*) or outside (see *Figure 2D*) the period of saccade execution. *Figure 2E* summarizes all five conditions tested in the experiment. Baseline sessions contained no context but only test trials. The remaining four sessions that did contain context trials, comprised a condition with no grating displacement and conditions with a displacement of the grating 37.89 ms (SEM 5.94 ms), 98.07 ms (SEM 8.59 ms) or 186.99 ms (SEM 7.92 ms) after saccade initiation.

After the presentation of the context trials, I tested displacement discrimination in a 2-alternative forced-choice task for a grating that was shifted either upwards or downwards (*Figure 2F*). The time-course of events was identical to that of the context trials except that the shift of the grating

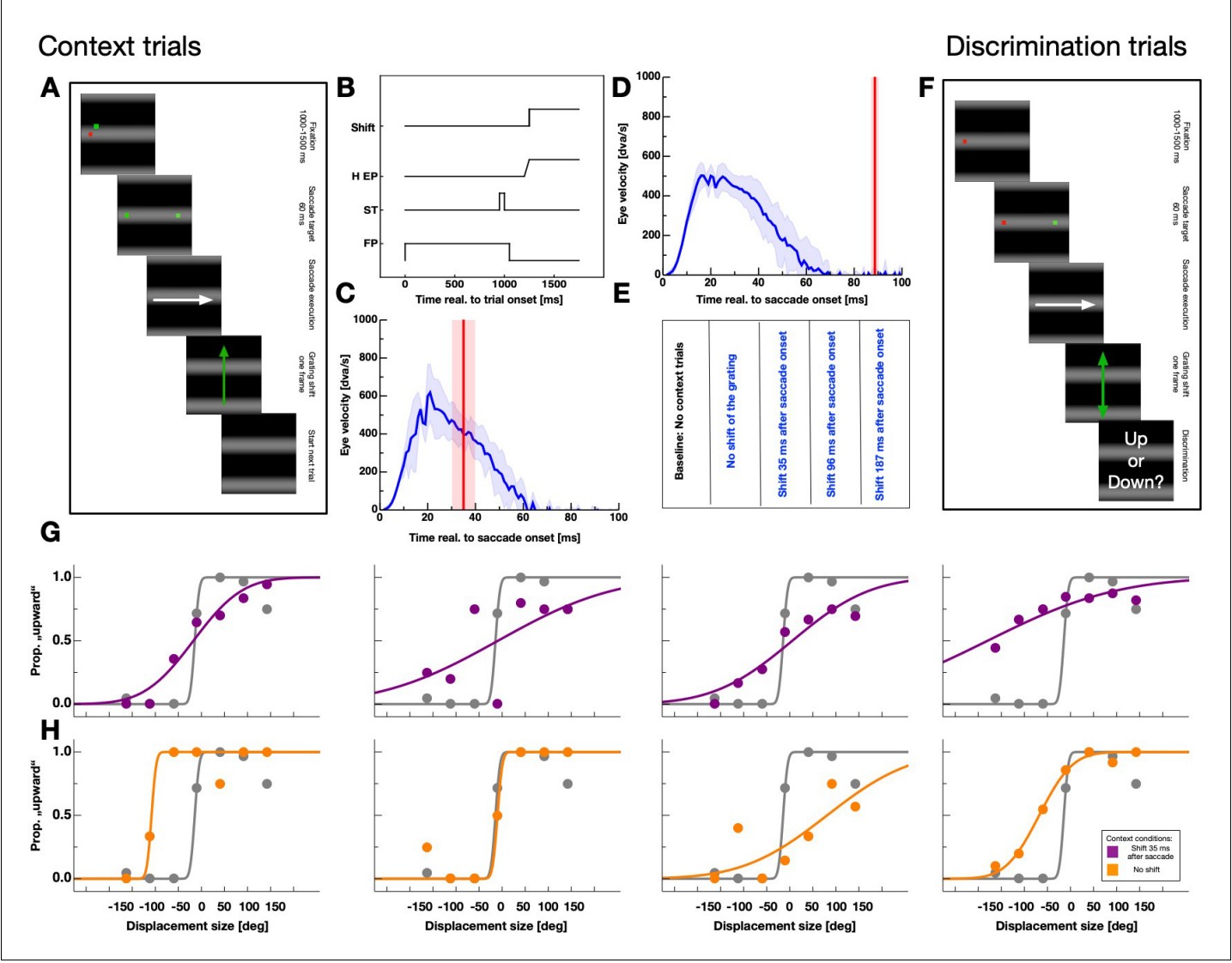

**Figure 2.** Experimental paradigm. (A) Schematic illustration of a context trial. A full-field grating was presented throughout the entire trial. A trial started with the presentation of a fixation point. After 1000–1500 ms a saccade target was presented together with the fixation point for 60 ms. Then, both, the saccade target and the fixation point disappeared. Subjects were instructed to perform a saccade to the remembered position of the saccade target as soon as the fixation point disappeared. Depending on the session, the grating was displaced upwards after 35, 98, 187 ms or not displaced at all. (B) Time-course of events in the context trials for the fixation point (FP), the saccade target (ST), the seven horizontal eye position (H EP) and the grating displacement (Motion). (C,D) Example eye velocity profiles from one participant representing saccade performance in the context trials (shown in blue). Average eye position is shown by the blue line. Average timing of the actual grating displacement is shown by the red vertical line and the standard deviation of the timing by the shaded area. (E) Summary of conditions. Baseline sessions consisted only of test trials. The remaining four sessions contained context and test trials. In these sessions the grating was displaced 35, 98, 187 ms after saccade execution or not displaced at all. (F) Schematic illustration of a displacement discrimination trial. These trials were identical to the context trials except that the grating was shifted upwards or downwards (indicated by the orange arrows) by various displacement sizes and across trials at various times relative to saccade onset. Participants were instructed to report the displacement direction at the end of the trial by pressing the corresponding arrow key on the computer keyboard. (G,H) Psychometric functions for judgements of the displacement direction in the test trials for all observers. Data in gray represent discrimination performance for grating displacements that occurred outside the saccade execution period and colored data peri-saccadic discrimination. Data in purple derive from sessions in which the grating displacement occurred 35 ms after saccade initiation in the context trials and data in orange from sessions with no grating displacement in the context trials. Discrimination performance was quantified by the JND of the psychometric function.

was applied at various times around saccade execution. I analyzed displacement discrimination performance relative to saccade onset in bins of 30 ms. *Figure 2G* shows displacement discrimination results of all participants from sessions in which the grating was shifted upward 35 ms after saccade onset in the context trials. Data were taken from a bin long before saccade initiation (<−50 ms to saccade onset, shown in gray) and from a bin close to saccade onset (−25 ms <saccade onset < 25 ms, shown in purple). Data were fitted by cumulative gaussian functions and performance determined by estimating the JND of the psychometric function.

Psychometric functions for grating displacements presented during the saccade were far shallower indicating higher thresholds or stronger suppression. Thus, suppression magnitude was strong when in the context trials the grating shifted upwards during saccade performance. Data shown in *Figure 2H* derive from sessions in which the grating was not displaced in the context trials. For all observers, JNDs of the psychometric functions from intra-saccadic bins were steeper than those measured after context trials in which the grating was displaced.

*Figure 3A* shows the full time course of displacement thresholds around the time of saccade execution for all observers. Data derive from sessions in which the grating was shifted 35 ms after saccade initiation. For all observers, intra-saccadic discrimination performance declined after they had performed saccades with intra-saccadic stimulation in the context trials. *Figure 3B* shows displacement thresholds from sessions where the grating was not shifted in the context trials. In order to quantify intra-saccade displacement discrimination, ie. suppression magnitude, I chose the peak thresholds that were contained in a bin ±50 ms around saccade initiation. For the quantification of displacement discrimination during fixation, I calculated the average over all bins lying at least 100 ms before and 100 ms after saccade onset.

I analyzed saccade parameters in order to estimate a putative influence of saccade amplitudes on suppression strength (*Volkmann et al., 1981*; *Stevenson et al., 1986*) in my data. *Figure 3D,E* shows horizontal and vertical average saccade amplitudes for all five sessions. In all sessions, saccades show the typical undershoot of the required amplitude (20°). Similarly, no statistical differences were found between average peak velocities of saccades (see *Figure 3C*). It has been shown that background displacement can provoke saccade adaptation (*Deubel, 1991*; *Robinson et al., 2000*; *Ditterich et al., 2000*). The recurring upward displacement of the background might have induced adaptation of the vertical saccade component. I subtracted the last ten vertical saccade amplitudes of the context trials by the first ten to check for an adaptation of saccadic gain. *Figure 3F* shows saccade gain change for all four sessions that contained context trials. I calculated one-tailed paired t-tests separately for all four sessions. There was no statistical evidence for a significant gain change in any of the sessions (context 35 ms: t(3) = 1.18, p=0.32, context 98 ms: t(3) = −0.60, p=0.59, context 187 ms: t(3) = −0.66, p=0.55, context no motion: t(3) = 0.21, p=0.85).

*Figure 3G* shows displacement discrimination averaged across all four participants from the peri-saccade period (shown in red) and the fixation period (shown in gray) in baseline sessions where no context trials were applied. Observers demonstrated poor peri-saccadic displacement discrimination as is expected if suppression is not absent. Gratings had to be shifted upwards on average ~120 deg (phase shift) such that observers were able to tell the correct direction of the shift. A paired one-tailed T-test confirmed that displacement discrimination thresholds were significantly higher when the grating was shifted peri-saccadically than before saccade initiation (t(3) = −9.38, p=0.0026). The effect size of this difference was 1.7 (Cohen's d) which is considered a very large effect. *Figure 3H* shows average displacement discrimination from sessions that included context trials. Intra-saccadic displacement discrimination thresholds were virtually identical to those of the baseline sessions if in the preceding context trials the grating was displaced during saccade execution, that is 35 ms after saccade onset. However, thresholds decreased massively after participants performed context trials in which the displacement occurred after the saccade had landed or in which there was no displacement at all. In these sessions, peri-saccade displacement discrimination was almost as good as during fixation. This finding provides clear evidence for the idea that the sensorimotor system habituates to peri-saccadic stimulation. In sessions where the grating was shifted outside the period of saccade execution, the system was relieved of suppression since no intra-saccadic stimulation occurred for more than 100 trials. For statistical analysis, I calculated the difference between baseline and adaptation sessions within each participant and averaged across all post-saccadic/no motion difference values (i.e. the 96, 187 ms and the No motion context data) within each

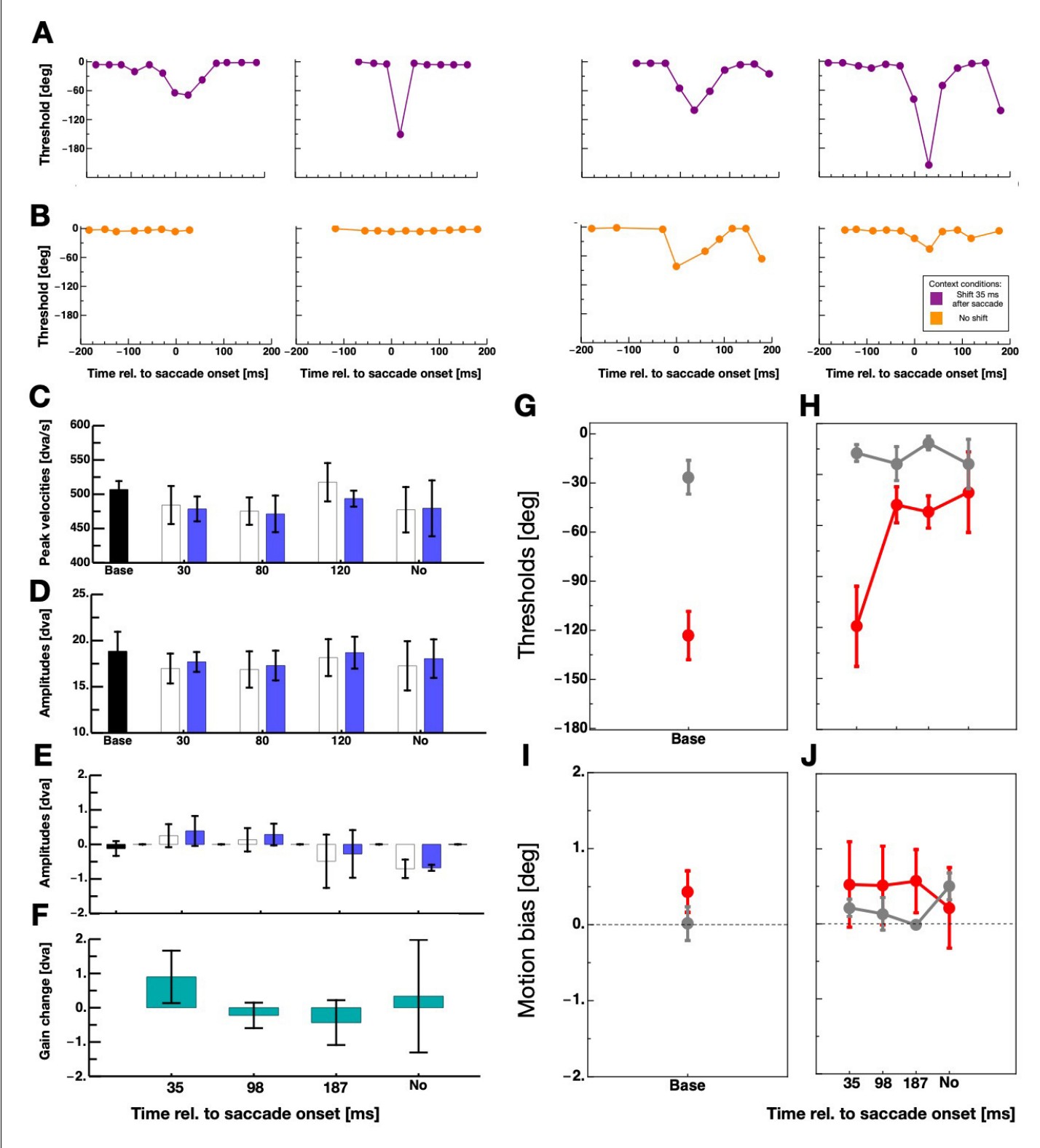

**Figure 3.** The effect of context on intra-saccadic motion discrimination. (A,B) Thresholds for perceiving grating displacements - measured in the test trials - as a function of time relative to saccade onset for all participants. Data shown in purple derive from sessions in which the grating was displaced 35 ms after saccade onset in the context trials and data shown in orange from sessions with no grating displacement in the context trials. (C) Average saccade peak velocities in the five sessions for saccades performed in context trials (white bars) and saccades performed in test trials (blue bars). The black bar indicates average data from the baseline sessions and the blue bars average data from the context sessions. Error bars represent S.E.M. (D)
*Figure 3 continued on next page*

*Figure 3 continued*

Average horizontal amplitudes in the five sessions for saccades performed in context trials (white bars) and saccades performed in test trials (blue bars). Same conventions as in 3C. (**E**) Average vertical amplitudes in the five sessions for saccades performed in context trials (white bars) and saccades performed in test trials (blue bars). Same conventions as in 3C. (**F**) Saccade gain change calculated by subtracting the last ten vertical saccade amplitude of the context trials by the first ten vertical saccade amplitude of the context trials. (**G**) Average displacement discrimination thresholds - measured in the test trials - from baseline sessions. Data in gray represent thresholds from discrimination performance measured outside the period of saccade execution and data in red performance measured around saccade execution. Error bars represent S.E.M. (**H**) Average displacement discrimination thresholds - measured in the test trials - that were preceded by context trials. Same conventions as in 3G. (**I**) Average motion bias - measured in the test trials - from baseline sessions. Same conventions as in 3G. (**J**) Average motion bias - measured in the test trials - from test trials that were preceded by context trials. Same conventions as in 3G.

participant. A paired one-tailed T-test confirmed that intra-saccadic stimulation in the context trials induced stronger suppression than post-saccadic/no displacement stimulation (t(3) = 3.68, p=0.017).

In order to better understand the effect of the intra-saccadic stimulation, I analyzed the biases of the displacement discrimination. *Figure 3I* shows biases for fixation (gray) and peri-saccade discrimination form the baseline sessions. A paired one-tailed T-test did not reveal a significant differences (t(3) = −1.63, p=0.20). Similarly in the discrimination data that followed context trials, no significant difference in biases between conditions was found (see *Figure 3J*). Thus, there was no evidence for a motion after-effect in the current data that would have manifested in changed bias, that is a shift of the psychometric functions.

Next, I sought to find out whether the habituation effect is direction-specific. To this end, I modified the display of the test trials to contain two horizontal gratings (see *Figure 4A*).

The experiment contained context and test trials. In the test trials, participants were required to perform a saccade to the remembered position of the saccade target as in the first experiment. Then, they had to judge which of the two gratings was displaced. The displacement direction in the test trials was always upwards and applied either during saccade or after saccade execution. The displacement direction in the context trials was, in separate sessions, either upwards or downwards with the same displacement size of 57° phase shift as in the other experiments. A clear elevation of discrimination thresholds was observed only if the displacement in the context trials occurred during the saccade and had the same direction (upwards) as in the test trials (see *Figure 4B*). A non-parametric repeated measures ANOVA with the factors 'displacement direction' (upward/downward) and 'displacement time' (during/after saccade) revealed a significant main effect 'displacement direction' (F(1,4) = 15.65, p=0.017), confirming that thresholds were elevated selectively when the displacement direction in the context and the test trials matched. The significant interaction effect (F(1,4) = 8.18, p=0.046) revealed that the selectivity was observed only when the displacement occurred during but not after saccade execution. The factor 'displacement time' was not significant (F(1,4) = 6.668, p=0.061).

Displacing a stimulus during execution of a saccade produces a different retinal motion vector as a displacement during ocular fixation. The retinal motion vector is the sum of the physical displacement vector (i.e. upward) and the inverse saccade vector (i.e. leftward for rightward saccades). Even though the screen borders were covered by a foil and the room was dark, the vertical edges of the horizontal grating might have contributed to the experience of a horizontal motion component. In order to compare habituation for intra-saccadic and post-saccadic presentations with a matched retinal motion vector, I conducted a new experiment. In this Experiment 3, I added vertical bars to the horizontal grating to imitate leftward motion of the vertical grating edges. The spatial frequency of the vertical bars and the horizontal grating was kept identical. The procedure was identical to Experiment 2 except that only in the test trials the displacement now contained a horizontal component in addition to the vertical. The amount of the horizontal component was calculated to be identical to the saccade-induced horizontal shift of the grating displacement measured in Experiment 2 (see Materials and methods section for details). I tested sessions with an upward displacement in the context trials (see *Figure 5B*) and a downward displacement (see *Figure 5C*). In the test trials the physical displacement was either upward or downward when presented in the intra-saccadic period or it was upward + leftward or downward + leftward when presented in the post-saccadic period. As in Experiment 2, habituation was much stronger in the intra-saccadic period and selective for the direction of the displacement.

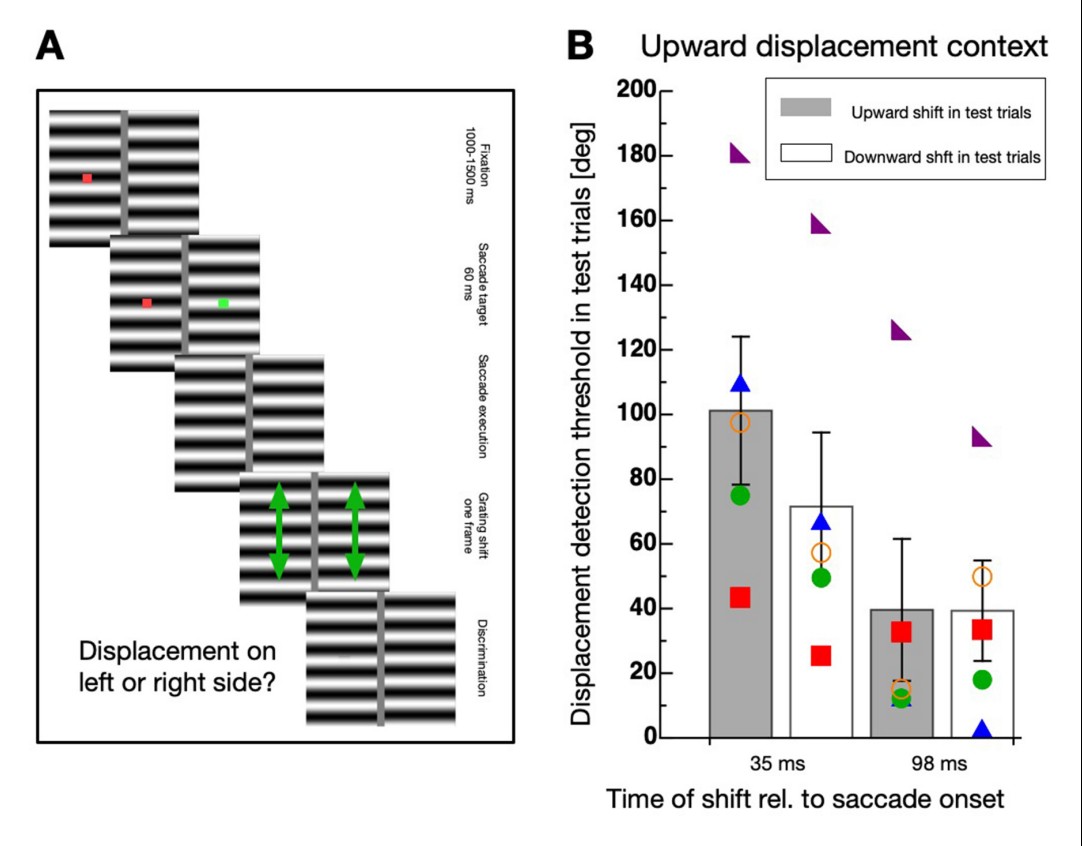

**Figure 4.** Direction-specificity of intra-saccadic habituation. (**A**) Schematic illustration of a displacement discrimination trial testing direction-specificity of the physical displacement direction. Two gratings were presented, one on the left and one on the right side of the screen. At trial start a fixation point (red color) was shown for 1000–1500 ms, then a saccade target (green color) appeared. After 60 ms both, the fixation point and the saccade target disappeared and participants were required to perform a saccade to the remembered position of the saccade target. One of the two gratings was displaced either during the saccade or after the saccade in either upward or downward direction. At the end of the trials participants had to indicate whether a displacement was seen at the left or the right side of the screen by pressing the corresponding arrow key on the computer keyboard. (**B**) Average displacement discrimination thresholds measured in the test trials. Bars shown in gray display average thresholds after upward displacements occurred in the test trials and white bars show thresholds after downward displacements occurred in the test trials. The small colored objects indicate performance of individual participants. Error bars represent S.E.M.

A non-parametric repeated measures ANOVA was calculated separately for data deriving from sessions with upward displacements in the context trials (see *Figure 5B*) and from those including downward displacements (see *Figure 5C*). For contexts with upward displacements, a significant main effect 'displacement time' ($F_{(1,4)}$ = 65.556, p<0.001) and a significant interaction effect ($F_{(1,4)}$ = 19.883, p=0.011) was revealed, confirming that the selective decrease of discrimination occurred during but not after saccade performance. The factor 'displacement direction' was not significant ($F_{(1,4)}$ = 5.00, p=0.089). For contexts with downward displacements, a significant main effect 'displacement time' ($F_{(1,4)}$ = 16.337, p=0.016) and a significant interaction effect ($F_{(1,4)}$ = 25.600, p=0.007) was revealed, confirming that the selective decrease of discrimination occurred during but not after saccade performance. The factor 'displacement direction' was not significant ($F_{(1,4)}$ = 4.986, p=0.089).

## Discussion

The results of this study clearly demonstrate that saccade suppression magnitude is strongly modulated by the recent context of intra-saccadic stimulation. After hundreds of saccades without intra-saccadic stimulation, observers were able to detect intra-saccadic displacements almost as good as

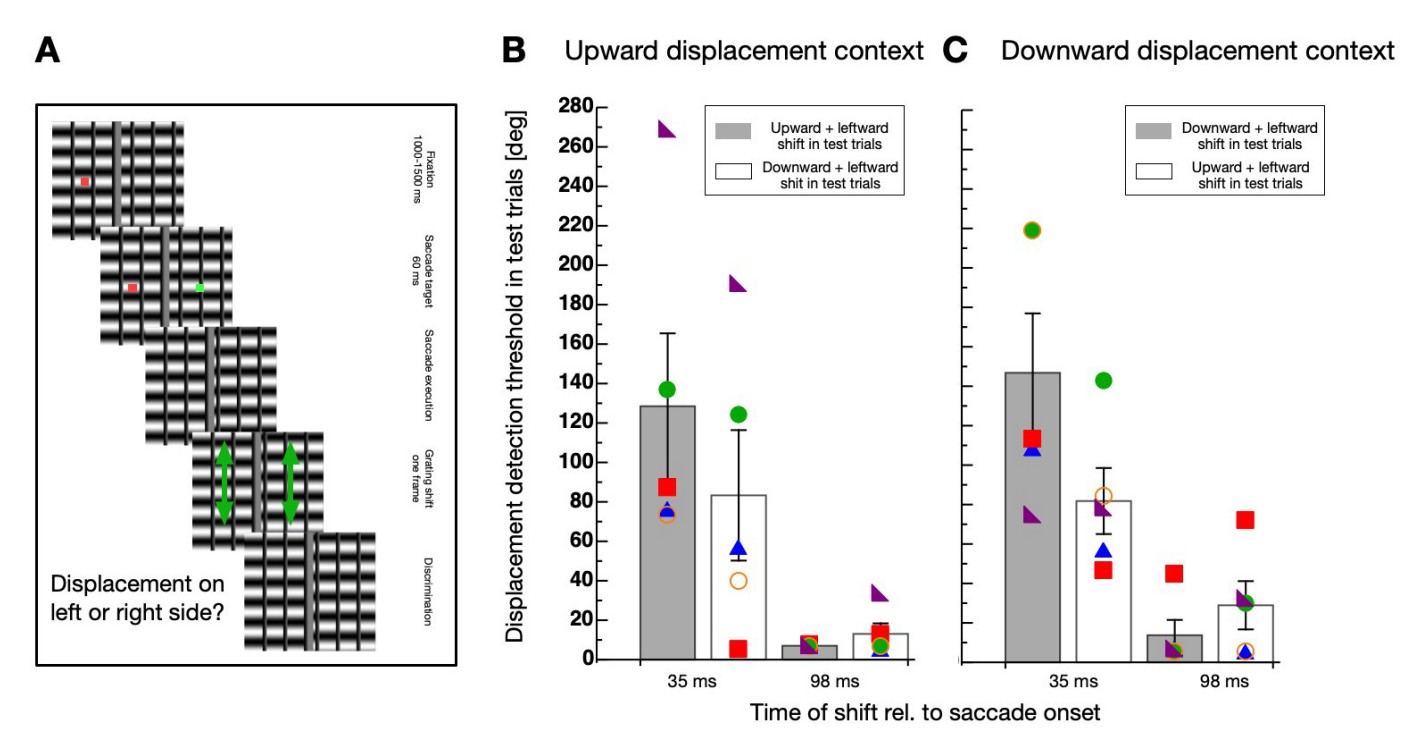

**Figure 5.** Direction-specificity controlled for retinal speed. (A) Schematic illustration of a displacement discrimination trial testing direction-specificity of the retinal motion direction. The procedure is identical to that described in *Figure 4A* except that different gratings were used in the test trials that were displaced in upward and leftward direction. (B) Average displacement discrimination thresholds measured in the test trials in sessions containing an upward motion context. Bars shown in gray display average thresholds after upward displacements occurred in the test trials and white bars show thresholds after downward displacements occurred in the test trials. The small colored objects indicate performance of individual participants. Error bars represent S.E.M. (C) Average displacement discrimination thresholds measured in the test trials in sessions containing a downward displacement context. Same conventions as in B).

during fixation. However, if the displacement was presented systematically within the period of saccade execution, thresholds increased drastically. The visual system habituates context-dependent to saccade-contingent visual information. This results provides evidence for a possible mechanism of peri-saccadic suppression. Instead of a general shut-down of vision, neurons might saturate to the information most prevalent in the previous saccadic context and thereby cancel out disturbing intra-saccadic stimulation. Importantly, any visual adaptation induced was contingent on the execution of a saccade. No habituation effect was found when the stimulation in the context trials was presented after a saccade had finished. The context dependent habituation therefore requires an extra-retinal signal that informs visual areas about the upcoming saccade. Such a context-dependence would be consistent with a recent Kalman filter model of saccade suppression in which higher variance in sensory feedback yields a reduced sensory weight, thereby decreasing sensitivity (*Crevecoeur and Kording, 2017*). In the model, the strength of saccade suppression is dynamically linked to the predicted reliability of the intra-saccadic sensory signal.

However, from the present data it cannot be determined how stimulus-specific trans-saccadic habituation is. Although I found direction-selectivity, it cannot be ruled out that habituation would also occur for changes of form or any visual stimulus occurring transiently during the saccade. One plausible candidate for the habituation to trans-sacadic displacements might be saturation of motion-sensitive neurons. Saturation of neuronal responses after short-term motion adaptation has been reported to occur in area MT (*Priebe et al., 2002*). The idea of the extra-retinal storage (see *Figure 1*) does not exclude however, that habituation occurs for any stimulus feature that reoccurs during saccade performance. This question should be pursued in future research by testing whether habituation is selective for trans-saccadic motion stimuli or if it generalizes across other visual features. The experimental task that I applied can in principle be solved without relying on motion

information. The direction of a displacement can be detected by two ways: By noticing the motion transient between the first and the second image or if that is not possible by comparing feature locations of the first and the second image. Displacement detection is strong during ocular fixation (*Legge and Campbell, 1981*) but weak during saccades (*Bridgeman et al., 1975*; *Deubel et al., 1996*; *Niemeier et al., 2003*). In order to compare the results of the current study to the phenomenon of suppression of displacement, one should look at studies that displaced the saccade target constantly in the same direction before testing displacement detection. This was done when the effect of saccade adaptation on displacement detection was tested (*Collins et al., 2009*). In this study, no effect of saccade adaptation trials on the displacement thresholds was observed (see their Figure 3C). However what they did report was a shift of the bias in displacement detection after saccade adaptation. In my study however, the bias of displacement discrimination was statistically indistinguishable before and after the context trials. In the absence of a conscious displacement detection, the oculomotor system might be aware of the shift and - in the case of systematic displacements - adapt to it. Some studies have shown that saccades not only adapt to intra-saccadic jumps of the saccade target but also to displacements of the background (*Deubel, 1991*; *Robinson et al., 2000*; *Ditterich et al., 2000*). I compared the first and the last 10 saccade amplitudes of the context trials and did not find significant changes that would indicate adaptation. The most likely reason for the absence of adaptation is that the previous studies used background images containing focal elements that allow estimating the distance to the eye position. An increase of visual sensitivity starting around 50 ms before saccade onset can be observed at the focus of attention, that is the saccade target (*Rolfs and Carrasco, 2012*). However, I used a low-frequency sinusoidal grating without clear edge information. In the experiments of the present study, it cannot be decided to what extent participants employed a comparison between the pre-and post-saccadic image and how much they used the motion transient to judge the displacement. In real life we constantly perform saccades of various sizes and directions, producing different retinal motion stimulations. The direction-selectivity that I found suggests that different saccade vectors are connected to habituation of different displacement directions. If there would a general insensitivity to displacements, direction-selectivity could not have been observed.

It has been suggested that suppression of trans-saccadic displacements is determined by peri-saccadic suppression of contrast sensitivity. In this view, an active mechanism transiently shuts down the magnocellular pathway and thereby prevents an intra-saccadic motion sensation (*Burr et al., 1994*). Others have demonstrated that some intra-saccadic motion perception is possible if stimuli are made optimal to compensate for the retinal displacement (*Castet and Masson, 2000*). Testing whether suppression of visual sensitivity is context-sensitive as well could answer the question of a link between peri-saccadic contrast sensitivity and displacement discrimination directly. Under the assumption of such a linkage, performing a number of saccades on a background with high contrast (even without motion information) should increase peri-saccadic displacement thresholds. However, the direction selectivity that I found indicates already that the habituation cannot be explained by the decrease in contrast sensitivity which would affect all displacement directions likewise.

Peri-saccadic increase in contrast thresholds might underlie a similar habituation mechanism. It has been repeatedly reported that contrast suppression (*Brooks and Fuchs, 1975*; *Brooks et al., 1980*; *Mitrani et al., 1971*; *Richards, 1969*) and the neural modulations in area MT (*Price et al., 2005*) are absent when tested under dark conditions. Consistently, intra-saccadic motion becomes visible if the light is switched on only during execution of the saccade (*Campbell and Wurtz, 1978*). However, suppression remains strong when saccades are made under ocular diffuser or in a Ganzfeld (*Riggs and Manning, 1982*). Since the peri-saccadic stimulation is identical in the dark and the Ganzfeld and only the illumination differs, the differences between suppression under both conditions cannot be attributed to backward masking. The visual system can habituate to intra-saccadic motion only under light conditions, thus probably explaining the absence of suppression in the dark. If suppression of visual contrast depends on context, performing a number of saccades in the Ganzfeld should release habituation and thereby increase intra-saccadic contrast sensitivity.

In conclusion, the data of the present study indicate that saccade suppression is context dependent. The experiments suggest a simple mechanism that explains how the disturbing intra-saccadic experience is prohibited by saccade-contingent neuronal habituation.

# Materials and methods

## Apparatus

Subjects were seated 45 cm from a Eizo FlexScan T57S with the head stabilized by a forehead rest. The visible screen diagonal was 20 in., resulting in a visual field of 40 dva × 30 dva. Stimuli were presented on the monitor with a vertical frequency of 120 Hz on a homogeneous gray background.

## Participants

Four subjects participated in the context-selectivity experiment (2 female, 2 male, mean age: 28; 3 naive to the purpose of each condition and the author), five subjects in Experiment 2 (3 female, 2 male, mean age: 29; 4 different subjects than in Experiment 1 and naive to the purpose of each condition and the author) and five subjects in Experiment 3 (4 female, 1 male, mean age: 29; 3 different subjects than in Experiment 1 and naive to the purpose of each condition, one subject that already participated in Experiment 2 and the author). All subjects had normal or corrected-to-normal vision. Experimental procedures were approved the local ethics committee of the psychological department of the Heinrich- Heine University Düsseldorf. Written informed consent was obtained prior to each experiment in accordance with the declaration of Helsinki.

## Eye movement recording

Eye movements were monitored by the EyeLink 1000 system (SR Research), which samples gaze positions with a frequency of 1000 Hz. Viewing was binocular, but only the dominant eye was recorded. A standard 9 point calibration was performed at the beginning of each block of trials. The system detected the start and the end of a saccade when eye velocity exceeded or fell below 22 dva/s and acceleration was above or below 4000 dva/s$^2$ respectively.

## Timing of stimulus presentation

In order to test the exact time of stimulus presentation on the screen, I used a photodiode that was connected to an electronic circuit and sent a TTL impulse to the parallel port of the eyelink PC for every luminance change on the screen. A lowpass filter prohibited that the luminance change at the time of the screen refresh entered into signal recording. This measuring device was custom built with electronic parts. The delay of the device itself was measured by an oscilloscope and found to operate in the nanosecond range. Thus, the measuring device plus the recording in the eyelink PC should not have added any detectable delay. In the analysis, I compared the timestamp of the message that my stimulus program sent to the Eyelink PC against the timestamp of the TTL impulse sent by the measuring device. With the setup that I used this display lag was measured to be on average 4.35 ms (S.E.M. 0.29). Therefore, I added 4.35 ms to all data in the analysis.

## Trial structure and data analysis

In all experiments, a session contained 385 trials. Experiment 1 consisted of 5 different types of session. Four of these sessions each contained context trials in which motion was displayed relative to the saccade (grating moved 35/98/187 ms after saccade onset) or no motion was displayed (grating remained stationary). Importantly, in one session only one type of context trials was presented. The fifth type of sessions were baseline sessions which did not contain context trials. In order to obtain sufficient data for the estimation of the suppression curve, participants repeated sessions of each context type several times (context 35 ms: 8 (S.E. 1.96), context 98 ms: 6 (S.E. 1.22), context 187 ms: 6.5 (S.E. 0.96), context no motion: 4 (S.E. 0.91), baseline: 7.75 (S.E. 0.75)). The order in which participants completed these sessions was randomized. The required saccade distance in each trial of all experiments was 20 dva. Trials in which saccade amplitudes were shorter than 10 dva were excluded from analysis.

For statistical analysis of Experiments 2 and 3, a non-parametric repeated measures ANOVA was calculated, using the Aligned Rank Transform (*Wobbrock et al., 2011*). Significance was determined by applying the Kenward-Roger approximation to estimate p-values, a procedure that has been shown to produce acceptable Type 1 error rates even for smaller samples (*Luke, 2017*).

## Procedure

All experiments were carried out in a complete dark environment. To avoid visibility of the screen borders, the display was covered with a transparent foil that reduced the luminance by about 2 log units. Each experimental session, except baseline sessions, contained context trials and test trials (see *Figure 1E*). Context sessions began with 105 context trials. After these, 5 test trials alternated with 5 context trials until the end of the session, containing 385 trials in total.

In Experiment 1, a context trial started with the presentation of a horizontal sinusoidal grating (spatial frequency: 0.05 c/dva, Michelson-contrast: 0.56, mean luminance: 0.15 cd/m2) that was displayed full-screen (see *Figure 1A*). On top of the grating a fixation rectangle (green, 0.75 dva x 0.75 dva) was shown 10 dva to the left of screen centre at the horizontal meridian. After a randomly chosen period between 1000 and 1500 ms, a saccade target (green, 0.75 dva x 0.75 dva) was shown while the fixation rectangle remained visible. After 60 ms, both, fixation rectangle and saccade target disappeared and participants performed a rightward saccade to the remembered position of the target. In all sessions rightward saccades were tested. The detection of the saccade by the eye-tracker triggered the displacement of the grating. Three different grating displacement times were programmed that were applied in separate sessions: The physical displacement occurred on average either 34.89 ms (SEM 5.94 ms) with a saccade duration of 58.67 ms (SEM 4.86 ms), 98.07 ms (SEM 8.59 ms) with a saccade duration of 52.32 ms (SEM 7.94 ms) or 186.99 ms (SEM 7.92 ms) with a saccade duration of 64.12 ms (SEM 6.23 ms) after saccade initiation. The grating displacement size in context trials was always 57 deg phase shift for one frame in upward direction. When participants had finished their saccade, they pressed one of the arrow keys to start the next trial. Test trials were identical to context trials except that the fixation point turned red and the intra-saccadic displacement of the grating could be upwards or downwards with one out of 7 possible displacement sizes (−115˚, −76˚, −38˚, 0˚, 38˚, 76˚, 115˚ phase shift). Each displacement size was presented 10 times in pseudo-randomized order. Participants were instructed to discriminate between upward and downward displacements by pressing the corresponding arrow key. After they pressed the response button a new trial started and participants re-directed their gaze to the fixation rectangle. During the leftward saccade to the fixation rectangle the grating remained stationary. Baseline sessions contained only test trials. From these data psychometric functions were determined by fitting cumulative gaussian functions to the motion discrimination data from each observer.

In Experiment 2, I measured the direction-selectivity of contextual habituation. All sessions in these experiments contained context and test trials. In context trials a full-screen grating (spatial frequency: 0.2 c/dva, Michelson-contrast: 0.56, mean luminance: 0.15 cd/m2) was presented that was - depending on session - displaced either during (i.e. 35 ms after saccade onset) or after (i.e. 98 ms after saccade onset) saccade execution. Each session contained 105 context trials after which 5 test trials alternated with 5 context trials. In test trials two gratings (each: spatial frequency: 0.2 c/dva, Michelson-contrast: 0.56, mean luminance: 0.15 cd/m2) were shown, one on each side of the screen (see *Figure 3A*). At trial start, a fixation point was shown for 1000–1500 ms. Then, a saccade target appeared for 60 ms after which both, the fixation point and the saccade target disappeared. The participants were instructed to perform a saccade to the remembered position of the saccade target as soon as it had disappeared. In test trials, one of the two gratings was displaced as soon as the eye-tracker detected saccade onset. The physical displacement occurred - depending on session - on average either 32.53 ms (SEM 0.94 ms) with a saccade duration of 52.8 ms (SEM 3.45 ms) after saccade initiation or 97.22 ms (SEM 0.78 ms) after saccade initiation with a saccade duration of 50.85 ms (SEM 4.66 ms). The displacement was chosen randomly across trials out of 7 possible sizes (−115˚, −76˚, −38˚, 0˚, 38˚, 76˚, 115˚ phase shift). Participants were instructed to indicate which of the two gratings was moved by pressing either the left or right arrow key.

A direct comparison of contextual habituation between intra-saccadic and post-saccadic motion discrimination is hindered by the fact that in intra-saccadic test trials the displacement is shifted on the retina by the saccade whereas in post-saccadic trials it is not. In Experiment 3, I tested the direction-selectivity of contextual habituation by matching the retinal motion vector imposed in intra-saccadic displacements with the motion vector in post-saccadic displacements. In Experiment 1 and 2 the physical displacement was vertical. For intra-saccadic displacements, the motion vector is the sum of the physical displacement direction and the direction of the retinal movement itself (i.e. leftward motion for a rightward saccade).

Experiment 3 was identical to Experiment 2, except that the grating displacement direction in the test trials was different. In order to present an oblique motion vector in the test trials, I now used a grating that contained vertical and horizontal bars (see *Figure 4*). The physical displacement occurred - depending on session - on average either 34.58 ms (SEM 0.78 ms) with a saccade duration of 67.8 ms (SEM 4.42 ms) after saccade initiation or 94.07 ms (SEM 4.06 ms) after saccade initiation with a saccade duration of 68.95 ms (SEM 3.56 ms). As the aim of Experiment 3 was to imitate the displacement of the horizontal grating edge produced by the saccade, one single bar could have been presented. However, one single bar would have served as an ideal landmark for localization. To prevent the use of such a landmark, I used the same spatial frequency for the generation of the vertical and the horizontal bars (0.2 c/dva). These vertical bars were generated by a sinusoidal function (each: spatial frequency: 0.2 c/dva, Michelson-contrast: 0.56, mean luminance: 0.15 cd/m2) but only a quarter of a cycle around the peak of the minimum luminance was shown (i.e. the dark bar). To estimate the displacement size that would mimic the retinal shift produced by the saccades, I used the average data of Experiment 2 and calculated the average eye velocity at the time of the displacements. To this end, I selected eye position samples that were recorded within 8 ms (i.e. one frame) after the time of the physical grating displacement. With these samples, I calculated the eye velocity at the frame that the displacement occurred ($v_{eye}$ = 181.25 dva/s). Then, I computed the necessary temporal frequency ($f_T$) of the grating that would mimic the eye velocity at the given frame rate (fps = 120 Hz) and the given spatial frequency of the grating ($f_S$ = 0.2 c/dva) by solving the equation: $f_T/f_S = _{veye}$ (36.25 c/s / 0.2 c/dva = 181.25 dva/s), dividing by the frame rate (36.25 c/s / 120 = 0.302 c/fps), i.e 0.302 × 2 x $\pi$ = 1.89 radians or 108.29 degrees.

## Acknowledgements

This research was supported by the Deutsche Forschungsgemeinschaft, DFG (ZI 1456) and by the European Research Council (project moreSense grant agreement n. 757184).

## Additional information

### Funding

| Funder | Grant reference number | Author |
|---|---|---|
| Deutsche Forschungsge-meinschaft | ZI/1456 | Eckart Zimmermann |
| H2020 European Research Council | 757184 | Eckart Zimmermann |

The funders supported the current study.

### Author contributions

Eckart Zimmermann, Conceptualization, Software, Formal analysis, Funding acquisition, Validation, Investigation, Visualization, Methodology

### Author ORCIDs

Eckart Zimmermann https://orcid.org/0000-0002-1964-2711

### Ethics

Human subjects: The study was approved by the ethics committee of the Faculty of Mathematics and Natural Sciences of the Heinrich-Heine-University Duesseldorf (ZI01-2019-01). Written informed consent about pseudonymized data collection, storage and publication was obtained prior to each experiment in accordance with the declaration of Helsinki.

### Decision letter and Author response

Decision letter https://doi.org/10.7554/eLife.49700.sa1
Author response https://doi.org/10.7554/eLife.49700.sa2

## Additional files

### Supplementary files

• Transparent reporting form

### Data availability

All measured data have been deposited in OSF at https://doi.org/10.17605/OSF.IO/TZA7F.

The following dataset was generated:

| Author(s) | Year | Dataset title | Dataset URL | Database and Identifier |
|---|---|---|---|---|
| Zimmermann E | 2019 | Saccade suppression depends on context | https://doi.org/10.17605/OSF.IO/TZA7F | Open Science Framework, 10.17605/OSF.IO/TZA7F |

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
