## [Decision Letter]

Thank you for submitting your article "Saccade suppression of motion depends on context" for consideration by *eLife*. Your article has been reviewed by three peer reviewers, and the evaluation has been overseen by a Reviewing Editor and Floris de Lange as the Senior Editor. The following individuals involved in review of your submission have agreed to reveal their identity: Martin Rolfs (Reviewer #2); Therese Collins (Reviewer #3).

The reviewers have discussed the reviews with one another and the Reviewing Editor has drafted this decision to help you prepare a revised submission.

Summary:

In this study, the author investigated the impact of recent visual experience on suppression of intra-saccadic visual stimulation. This is a timely study and the idea is intriguing, the experiments are well designed, and the results seem robust. But some major and other theoretical concerns should be addressed. The reviewers suggest an experiment that should be conducted which would address some major concerns. In addition, many methodological and data analysis issues should be stated and/or clarified. Lastly, a suggested additional analysis would strengthen the manuscript.

Essential revisions:

1) It is not clear that the author studies motion perception at all. Instead, the author uses a displacement task, which seems to rely on position rather than motion information. Indeed, the author might not be studying the perception and suppression of motion (as studied by Burr, Castet, and others), but of displacement (i.e., Bridgeman, Hendry, and Stark, 1975; Niemeier, Crawford, and Tweed, 2003). Both processes have been associated with rather different theoretical implications, and with an emphasis on different aspects of visual stability. Indeed, it seems that the study says more about the visual system's ability to disregard displacements of the retinal image across the saccade rather than about the perceptual omission of intra-saccadic motion. In my view, this does not reduce the novelty of the approach, but leads to a rather different interpretation of the meaning of these results with respect to understanding visual stability.

2) There are insufficient links to previous work about motion perception during saccades. In the Abstract, the author states that "…the mechanism that inhibits intra-saccadic motion detection is unknown". The author cites only Castet and Masson, 2000, but further work from Eric Castet and colleagues should be consulted: Castet, Jeanan and Masson, 2002; Duyck, Wexler, Castet and Collins, 2018, as well as Duyck, Collins and Wexler, 2016. These studies show that visual masking is a powerful determinant of saccadic suppression (as did Campbell and Wurtz, 1978). Thus, although some aspects of saccadic suppression of motion do remain to be fully described and uncovered, it is a misrepresentation of the literature to state that they are unknown.

Indeed, there is a dominant view/theory in the field: that the mechanism of reduced motion sensitivity is the peri-saccadic suppression of the magnocellular pathway, instantiated via efference-copy signals at the earliest stages of visual processing (Diamond et al., 2000; Thier et al; Burr et al., 1994; Burr and Ross, 1982; reviewed in Binda and Morrone, 2018). It seems that the present manuscript provides an alternative account to this dominant view and that should be emphasized here.

3) The article would also benefit from a clearer explanation of the difference between what Burr et al., 1994, showed, which is that there is a small decrease in contrast sensitivity (which they hypothesized might contribute to decreased motion sensitivity during saccades, but which they did not test); and what Castet and colleagues showed, which is that suppression of motion during saccades results not from this decrease of contrast sensitivity but from masking of static images before and after saccades.

4) Specify what other areas in addition to intra-parietal cortex shift their receptive fields predictively. There should be a more explicit link between the sentence of shifting receptive fields (Introduction) and the following sentence dealing with suppression of intra-saccadic motion perception. Additionally, it would be good to mention if some of the areas in which receptive fields shift predictively are likely to play a role in suppression of intra-saccadic motion perception.

5) The author should provide a clearer description of the model that he has in mind to account for reduced visibility of motion/displacement during saccades. In the Introduction, he states: "In this view, motion-sensitive neurons saturate their response to the visual information most dominant in the previous history of intra-saccadic stimulation." The idea expressed here is extended in the Discussion, but it needs further explanation. The author suggests that the system stores information (e.g. due to habituation) about motion that was recently experienced, but how could that motion be exclusive to saccades? Are the motion- (or displacement-) sensitive neurons that undergo habituation tuned to extremely high speeds? Or is the habituation linked to an efference-copy signal, forming a sensorimotor contingency? Perhaps visualize this with a schematic diagram.

6) The author states that saccade suppression starts 50 ms before saccade onset (Introduction). Interestingly, during this timing presaccade attention enhances perception (e.g. Kowler et al., 1995; Deubel and Schneider, 1996; Rolfs and Carrasco, 2012; Li, Barbot and Carrasco, 2016; Szinte, Jonikaitis, Rangelov and Deubel, 2018). What are your thoughts regarding whether and how these two phenomena may interact?

7) The author suggests a saccade-contingent habituation to intra-saccadic motion. It would be good to specify what 'previous history' means here (Introduction). Also, how would previous history translate to real life in which we constantly make saccades in all directions?

Methods and Analysis

8) The Materials and methods section does not contain any information on data pre-processing and data analysis, and many important aspects of the experimental methods are not specified.

9) Why was the design of Experiment 2 unbalanced? In other words, why weren't there test trials with downward motion? The result would be more convincing with a balanced design showing that motion detection thresholds were elevated in downward motion test trials as well. This is particularly important given known differences between upwards and downwards motion perception (e.g., due to gravity). If robust, the results reported here should be valid for both motion directions.

10) It is not always obvious what one is looking at in the figures, and understanding requires flipping back and forth between the Materials and methods and Results. It would be more conducive to comprehension if some details were made available at the appropriate parts of the Results section. For example, the psychometric functions in Figure 1G-H: aside from the fact that the figure is too small for comfortable reading, it seems that there must be a mistake in the x-axis label. It currently reads "time rel. to saccade onset", but should probably be "motion direction in test trials".

Another example, in Figure 2F-H: it is unclear whether the x-axis refers to time relative to saccade onset in context or test trials. Unfortunately, the text and figure captions do not make it any clearer (especially since the text says that motion discrimination performance in the test trials was analyzed in bins of 30 ms, but the x-axis is not in bins of 30 ms).

11) The timing of the stimuli with respect to the saccade is very critical in these experiments and more information needs to be provided. The author suggests that stimuli targeted one of 3 different times relative to saccade onset. Did the author confirm accurate timing of the displacements offline? How did the detected onset of a saccade online compare to the saccade onset detected offline? The author should provide measurements of their stimulus timing (using a photodiode or similar) relative to this ground truth.

Schweitzer and Rolfs (2019; https://www.biorxiv.org/content/10.1101/693309v1) have recently provided a detailed description of the delays contributing here, many of which are often unknown and need to be measured in a particular setup.

12) The author states, "In my experiments the adapting stimulus was a single-frame displacement only, whose effectivity might have accumulated over the bunch of trials.” To assess whether this was the case, the author should analyze separately the first vs. the second halves of the experiments and compare them.

Additional experiment

13) Across the two experiments, the author tests the specificity of a reduction in saccadic suppression to presentations occurring (1) during the saccade and (2) in directions that are congruent with those experienced during the context trials. However, given that the vertical displacement occurs during the saccade and given that the grating has vertical boundaries at the edges of the screen (they must be there, however minimal they may be), the vector of motion on the retina is changed due to the movement of the retina itself. Thus, the stimulus' motion across the retina has a vertical (due to stimulus motion on the screen) as well as a horizontal component (in the direction opposite the saccade). As a consequence, the context trials in which the motion/displacement is shown during the saccade should habituate the system to this combined retinal motion direction. If, as the author argues, the habituation is direction-specific, the consequence of this habituation (higher suppression) would thus only show up during test trials in which the motion/displacement is shown during the saccade, as the motion in these trials is again of the same direction. It would not be expected to show up in conditions (or to a reduced extent, compatible with the data) in which the context trials showed vertical motion while the eyes were stationary, as motion/displacement during the test trials (with intra-saccadic tests) would no longer be congruent with that experienced during the context trials.

An experiment to test this would use two different sets of context-trials in which the retinal direction of the stimulus is matched between intra-saccadic and post-saccadic context conditions. Alternatively, the context trials could be kept the way they were done in Experiment 2, but now the test trials in both intra-saccadic and post-saccadic context conditions could have one of two retinal directions: the direction imposed on the retina during saccades (vertical + horizontal) or the direction imposed during fixation (purely vertical). These experiments would reveal if the habituation is indeed specific to saccades or if the specificity to saccades found in the first two experiments is merely a consequence of differences in retinal motion trajectories. Moreover, the experiment should be run with a balanced design; i.e., habituation to upwards and downwards motion should be compared.

[Editors' note: further revisions were suggested prior to acceptance, as described below.]

Thank you for re-submitting your article "Saccade suppression of motion depends on context" for consideration by *eLife*. Your article has been re-reviewed by three peer reviewers, and the evaluation has been overseen by a Reviewing Editor and Floris de Lange as the Senior Editor. The following individuals involved in review of your submission have agreed to reveal their identity: Martin Rolfs (Reviewer #2); Therese Collins (Reviewer #3).

Our decision has been reached after consultation among the reviewers and the action editor. The reviewers appreciate that you have added new experiments to address the concerns raised in the previous round. They think that the findings from Experiment 3 are convincing, but that the findings from the Supplementary experiment are problematic for your proposal that this study deals with motion discrimination rather than displacement discrimination. Based on these discussions and the individual reviews below, we are inviting you to resubmit your manuscript, we are proposing a way forward for you to consider.

Experiment 3 seems convincing because it tries to match saccade-induced and screen motion, uses a balanced design (both upwards and downwards context motions are tested), and provides converging evidence for the directional tuning of the contextual habituation effect. Thus, the results of Experiment 2 are unlikely to be a consequence of differences in retinal motion vectors between different context conditions.

In the supplementary experiment a full-field random pattern mask flashed during saccade execution diminishes performance (comparison of pre- and post-saccadic target locations). The author concludes that because the mask abolished the motion transient, it is the motion transient that drives performance.

However, there is a serious logical flaw in this experiment. The results seem to be inconsistent with the author's theoretical proposal; i.e., that saccadic omission results from habituation to the motion transients that accompany every saccade (and therefore there is no need for special trans-saccadic mechanisms as claimed by many in the literature). In support of this claim, Experiment 1 shows that exposing observers to artificial intra-saccadic motion degrades motion discrimination performance (relative to performance without exposure to artificial intra-saccadic motion). The supplementary experiment is essentially the same as Experiment 1, except that now the upwards motion is masked by a full-field random pattern (although it still remains to be shown whether such a pattern actually does mask motion during normal viewing, i.e. fixation). If habituation to intra-saccadic motion explains poor motion discrimination, then elimination of intra-saccadic motion should release discrimination performance. Performance should therefore improve or, at least, remain comparable to the no-exposure condition (because there is, of course, still the usual saccade-induced motion transient in the no-exposure condition). Instead, results show the reverse – performance gets worse – which cannot easily be reconciled with the theoretical claim the author is making. Even if we consider that the mask itself generates motion on the retina during the saccade, the vector of that motion is horizontal and should not –according to the author's proposal – affect vertical motion detection.

Indeed, in the rebuttal, the author argues that his results do speak to motion perception because detecting the displacement change across saccades requires visibility of the motion transient. In support of his argument, he cites Shiori and Cavanagh, 1989, who showed that sensitivity to the motion transient was abolished during saccades. But this doesn't make much sense: if the motion transient is not perceived during the saccade, how can it be used to drive displacement reports? He does not cite (but could have) experiments performed by Deubel and Schneider in which they showed that performance increases dramatically when there is a blank (that diminishes the motion transient) between pre- and post-saccadic targets. So it seems that there is evidence that diminishing the motion transient may actually help trans-saccadic location performance.

Previous work from the author and Sabine Born showed that it is possible to mimic poor performance in a fixation task in which a mask occurs between the two targets. However, being able to degrade performance in two different ways (either by presenting targets before and after a saccade, or by presenting them before and after a mask) does not mean that the same processes are involved.

An alternative interpretation of the results is that the intra-saccadic mask might induce uncertainty about the initial position of the grating, making the displacement harder to judge (rather than rendering the motion invisible); for instance, the masking pattern may have itself induced transients on the retina that caused the system to down-regulate any input during subsequent saccades. This would argue that the author's protocol does not study motion perception (as expressed in the title of the manuscript), but displacement detection.

To justify the conclusion that is the motion transient that drives performance, two further pieces of information would be needed. (1) Does such a full-field mask abolish a motion transient of an embedded target (in a fixation condition)? (2) Why is the invisibility of the motion transient during a saccade (as shown by Shiori and Cavanagh) not sufficient? The author would have to provide evidence that the manipulation does indeed reduce motion perception while leaving the ability to judge displacements intact.

In sum, the only condition in which performance increased after context trials was the context in which the author minimized changes on the retina during the saccade (no displacement context trials). It seems, therefore, that any visual changes during saccades result in insensitivity during subsequent saccades. The plasticity in intrasaccadic omission remains an interesting finding, but it deviates quite strongly from the claim in the title/Abstract/conclusion that this paper is about saccade suppression of motion. Thus, we invite you to provide an alternative interpretation to saccadic suppression of motion for your current findings. Note that this would entail a re-writing of the entire manuscript, theory included. Please note that in *eLife* it is not customary to have many review rounds; thus, if you decide to take this route please try to make a comprehensive and clear a case as possible.

Analysis

In several of the new results, the author conducted multiple t-tests rather than an ANOVA, which is not good statistical practice. Take as an example: in Experiment 2, the author tests for direction-specificity of the contextual habituation with four conditions resulting from the crossing of two factors, Direction of Motion in Test Trials (Upward/Downward) and Time of shift (intra-sacc: 35 ms/post-sacc: 98 ms). In the results he only presents t-test on the Direction Factor, independently for each time of shift. Because the t-test is significant in intra-sacc, and not in post-sacc, the author concludes that there is direction-specific intra-saccadic habituation. Of course, this conclusion only holds if there is a statistically reliable interaction between the two factors. Factorial designs should be analyzed with ANOVAs throughout, instead of t-tests. To respond to the sample size critique, the author could use a linear mixed model with p-values estimated by the methods explained by Luke (2017, Behav Research Methods), which apparently is less sensitive to sample size than ANOVAs and t-tests.

One issue that further complicates the interpretation of the results of the supplementary experiment is that it is difficult to have an idea of power. How big of a difference in slope can 4 subjects reveal (given the variability seen in the other experiments, and especially when several cannot be fit by a cumulative Gaussian)?

---

## [Author Response]

Essential revisions:1) It is not clear that the author studies motion perception at all. Instead, the author uses a displacement task, which seems to rely on position rather than motion information. Indeed, the author might not be studying the perception and suppression of motion (as studied by Burr, Castet, and others), but of displacement (i.e., Bridgeman, Hendry, and Stark, 1975; Niemeier, Crawford, and Tweed, 2003). Both processes have been associated with rather different theoretical implications, and with an emphasis on different aspects of visual stability. Indeed, it seems that the study says more about the visual system's ability to disregard displacements of the retinal image across the saccade rather than about the perceptual omission of intra-saccadic motion. In my view, this does not reduce the novelty of the approach, but leads to a rather different interpretation of the meaning of these results with respect to understanding visual stability.

I agree that this is an important issue. Since I used a displacement, a link to the SSD phenomenon seems likely. However, there are arguments that speak against such a connection. For this reason, I avoided this topic in my first version of the manuscript. However, now I have added a large part in the Discussion section.

The question here is – as the reviewer point out – whether observers in the displacement task rely on position rather than motion. In the SSD literature the poor trans-saccadic displacement detection is usually explained by a saccadic suppression of the motion transient. Consequently, if observers can rely only on position their discrimination performance drops down. The reviewers suggest that through the context trials in which there is no motion observers improve their ability to rely on position, i.e. to compare pre and post-saccadic locations.

In order to test the ability of observers to compare pre-and post-saccadic location, I conducted an extra experiment. I modified my Experiment 1 such that the displacement in the test trials contained a whole-field random pattern mask that was presented for 24 ms exactly at the time when the displacement occurred. The motion transient of the displacement was therefore covered by the mask. The mask was presented short enough to be only on screen during saccade execution. If observers would be able to compare pre-and post-saccadic locations in the absence of the motion transient, they should be able to solve this task. However, for most of the data, a psychometric function could not even be estimated because the discrimination performance was very poor. Therefore, I conclude that the motion transient is necessary for trans-saccadic discrimination performance and that my context trials modulate the visibility of the motion transient.

I now write:

“In principle, the experimental task that I applied can be solved without motion information. […] Together, these arguments let it seem very unlikely that observers compared pre- and post-saccadic locations of image parts.”

2) There are insufficient links to previous work about motion perception during saccades. In the Abstract, the author states that "…the mechanism that inhibits intra-saccadic motion detection is unknown". The author cites only Castet and Masson, 2000, but further work from Eric Castet and colleagues should be consulted: Castet, Jeanan and Masson, 2002; Duyck, Wexler, Castet and Collins, 2018, as well as Duyck, Collins and Wexler, 2016. These studies show that visual masking is a powerful determinant of saccadic suppression (as did Campbell and Wurtz, 1978). Thus, although some aspects of saccadic suppression of motion do remain to be fully described and uncovered, it is a misrepresentation of the literature to state that they are unknown.

I have now re-written large parts of the Introduction:

“As it has been shown that the visual system is able to detect high-speed motion (Burr et al., 1982) the question arises why we are not aware of the motion stimulation produced by a saccade. […] Recently, Duyck, Wexler, Castet and Collins, 2018, used simulated saccades and found motion perception to be much less salient in the presence of static objects.”

Indeed, there is a dominant view/theory in the field: that the mechanism of reduced motion sensitivity is the peri-saccadic suppression of the magnocellular pathway, instantiated via efference-copy signals at the earliest stages of visual processing (Diamond et al., 2000; Thier et al.; Burr et al., 1994; Burr and Ross, 1982; reviewed in Binda and Morrone, 2018). It seems that the present manuscript provides an alternative account to this dominant view and that should be emphasized here.

I thank the reviewers for giving me the chance to point out the novelty of my account. I now write:

“Here, I offer a novel explanation of suppression which states that the brain stores and habituates to sensorimotor contingencies. […] In this view, motion-sensitive neurons store saccade-induced visual stimulation and saturate their response to the visual information most dominant in the previous history of the last bunch of saccades that has been executed.”

3) The article would also benefit from a clearer explanation of the difference between what Burr et al., 1994, showed, which is that there is a small decrease in contrast sensitivity (which they hypothesized might contribute to decreased motion sensitivity during saccades, but which they did not test); and what Castet and colleagues showed, which is that suppression of motion during saccades results not from this decrease of contrast sensitivity but from masking of static images before and after saccades.

I now describe the differences between the approaches by Burr et al., 1994, and Castet more clearly in the Introduction. Please see my reply to the reviewers’ point 2.

Additionally, I now write in the Discussion:

“It is unclear if suppression of motion transients is determined by peri-saccadic suppression of contrast sensitivity. […] However, the motion direction selectivity that I found indicates already that the habituation cannot be explained by the decrease in contrast sensitivity which would affect all motion directions likewise.”

Please note that already in the first version of the manuscript I stated in the last sentence of this paragraph (last sentence of the above quoted paragraph) that the decrease in contrast sensitivity does not explain the unawareness of intra-saccadic motion. I hope the new version now is more explicit on this point.

4) Specify what other areas in addition to intra-parietal cortex shift their receptive fields predictively. There should be a more explicit link between the sentence of shifting receptive fields (Introduction) and the following sentence dealing with suppression of intra-saccadic motion perception. Additionally, it would be good to mention if some of the areas in which receptive fields shift predictively are likely to play a role in suppression of intra-saccadic motion perception.

This section was meant to introduce the concept of efference copy which is primarily discussed in the remapping literature. I did not mean to say that remapping might somehow be related to suppression. I now rewrote that section and I also specify the other areas in which receptive field shifts have been measured. This section now reads:

“Learning sensorimotor contingencies requires a signal that indicates the initiation of a movement. […] The suppression of intra-saccadic motion perception might be driven by such an extra-retinal signal likewise (for a review, see Binda and Morrone, 2018).”

5) The author should provide a clearer description of the model that he has in mind to account for reduced visibility of motion/displacement during saccades. In the Introduction, he states: "In this view, motion-sensitive neurons saturate their response to the visual information most dominant in the previous history of intra-saccadic stimulation." The idea expressed here is extended in the Discussion, but it needs further explanation. The author suggests that the system stores information (e.g. due to habituation) about motion that was recently experienced, but how could that motion be exclusive to saccades? Are the motion- (or displacement-) sensitive neurons that undergo habituation tuned to extremely high speeds? Or is the habituation linked to an efference-copy signal, forming a sensorimotor contingency? Perhaps visualize this with a schematic diagram.

Yes indeed, I propose that storage of the past intra-saccadic motion stimulation is saccade-contingent. Indeed, this storage is linked to an efference-copy signal and thereby exclusive to saccades. I have now created a schematic diagram (see Figure 1) explaining this idea, as the reviewer suggested.

6) The author states that saccade suppression starts 50 ms before saccade onset (Introduction). Interestingly, during this timing presaccade attention enhances perception (e.g. Kowler et al., 1995; Deubel and Schneider, 1996; Rolfs and Carrasco, 2012; Li, Barbot and Carrasco, 2016; Szinte, Jonikaitis, Rangelov and Deubel, 2018). What are your thoughts regarding whether and how these two phenomena may interact?

This is a challenging question as attention shifts (increase of contrast sensitivity (Rolfs and Carrasco, 2012)) and saccade suppression (decrease of contrast sensitivity) lead to opposite effects. Both processes can co-exist because the benefits of attention shifts are at a focal region around the attended target which is usually the saccade target (Kowler et al., 1995; Deubel and Schneider, 1996) or under special conditions an increased region around the saccade target (Szinte, Puntiroli and Deubel, 2019). Suppression is mostly studied with gratings or bars that fill a large region of the visual field that is not the focus of attention. Interestingly, focal light points indeed are not suppressed (Brook and Fuchs, 1975). One could go further and argue that the insensitivity of contrast for the unattended regions is a consequence of the enhancement at the attended region. However, there is so far no evidence for a causal link between both processes. I now write in the Introduction:

“An increase of visual sensitivity starting around 50 ms before saccade onset can be observed at the focus of attention, i.e. the saccade target (Rolfs and Carrasco, 2012).”

7) The author suggests a saccade-contingent habituation to intra-saccadic motion. It would be good to specify what 'previous history' means here (Introduction). Also, how would previous history translate to real life in which we constantly make saccades in all directions?

I now write “in the previous history of the last bunch of saccades that has been executed.”. In the Discussion, I now suggest that in real life different saccade directions must involve different habituations. This is suggested by the direction-selectivity of the habituation as I now write:

“In real life we constantly perform saccades of various sizes and directions, producing different motion stimulations. The direction-selectivity that I found suggests that different saccade vectors are connected to habitation of different motion directions. If there would a general insensitivity to motion, direction-selectivity could not have been observed.”

Methods and Analysis8) The Materials and methods section does not contain any information on data pre-processing and data analysis, and many important aspects of the experimental methods are not specified.

I have now added the missing information.

9) Why was the design of Experiment 2 unbalanced? In other words, why weren't there test trials with downward motion? The result would be more convincing with a balanced design showing that motion detection thresholds were elevated in downward motion test trials as well. This is particularly important given known differences between upwards and downwards motion perception (e.g., due to gravity). If robust, the results reported here should be valid for both motion directions.

I agree and have now included a balanced design in my new Experiment 3.

10) It is not always obvious what one is looking at in the figures, and understanding requires flipping back and forth between the Materials and methods and Results. It would be more conducive to comprehension if some details were made available at the appropriate parts of the Results section. For example, the psychometric functions in Figure 1G-H: aside from the fact that the figure is too small for comfortable reading, it seems that there must be a mistake in the x-axis label. It currently reads "time rel. to saccade onset", but should probably be "motion direction in test trials".Another example, in Figure 2F-H: it is unclear whether the x-axis refers to time relative to saccade onset in context or test trials. Unfortunately, the text and figure captions do not make it any clearer (especially since the text says that motion discrimination performance in the test trials was analyzed in bins of 30 ms, but the x-axis is not in bins of 30 ms).

Thanks. I have corrected these errors. I have also put more details into the description of the results and I hope that it is more comprehensive now.

11) The timing of the stimuli with respect to the saccade is very critical in these experiments and more information needs to be provided. The author suggests that stimuli targeted one of 3 different times relative to saccade onset. Did the author confirm accurate timing of the displacements offline? How did the detected onset of a saccade online compare to the saccade onset detected offline? The author should provide measurements of their stimulus timing (using a photodiode or similar) relative to this ground truth.Schweitzer and Rolfs (2019; https://www.biorxiv.org/content/10.1101/693309v1) have recently provided a detailed description of the delays contributing here, many of which are often unknown and need to be measured in a particular setup.

Yes, this is an important point. I have carefully measured the display lag, i.e. the duration between the command to draw the stimulus is sent from the stimulus program and the actual appearance of the stimulus on the screen. This was done the following way as described also in the new section “Timing of stimulus presentation”:

“Timing of stimulus presentation

In order to test the exact time of stimulus presentation on the screen, I used a photodiode that was connected to an electronic circuit and sent a TTL impulse to the parallel port of the eyelink PC for every luminance change on the screen. […] With the CRT monitor that I used this display lag was measured to be on average 4.35 ms (S.E.M. 0.29). All data in the analysis were shifted in time by 4.35 ms.”

12) The author states, "In my experiments the adapting stimulus was a single-frame displacement only, whose effectivity might have accumulated over the bunch of trials.” To assess whether this was the case, the author should analyze separately the first vs. the second halves of the experiments and compare them.

As the reviewer suggested, I split my data into two halves and looked at the suppression strength separately. I found that already in the first half, suppression was reduced in the sessions without motion in the context trials. The strength of this effect was nearly identical to the result obtained when analyzing the total data set. What I meant to say with the sentence above is that it is likely that the habituation grows over several experiences of intra-saccadic displacements. It will surely not just happen with a single trial only. As for saccade adaptation, it is a separate question how many trials are needed to induce the effect (in my case: the habituation). This should be tested by varying the amount t of context trials in the beginning of a test session. For now, I have dropped the sentence quoted by the reviewer.

Additional experiment13) Across the two experiments, the author tests the specificity of a reduction in saccadic suppression to presentations occurring (1) during the saccade and (2) in directions that are congruent with those experienced during the context trials. However, given that the vertical displacement occurs during the saccade and given that the grating has vertical boundaries at the edges of the screen (they must be there, however minimal they may be), the vector of motion on the retina is changed due to the movement of the retina itself. Thus, the stimulus' motion across the retina has a vertical (due to stimulus motion on the screen) as well as a horizontal component (in the direction opposite the saccade). As a consequence, the context trials in which the motion/displacement is shown during the saccade should habituate the system to this combined retinal motion direction. If, as the author argues, the habituation is direction-specific, the consequence of this habituation (higher suppression) would thus only show up during test trials in which the motion/displacement is shown during the saccade, as the motion in these trials is again of the same direction. It would not be expected to show up in conditions (or to a reduced extent, compatible with the data) in which the context trials showed vertical motion while the eyes were stationary, as motion/displacement during the test trials (with intra-saccadic tests) would no longer be congruent with that experienced during the context trials.An experiment to test this would use two different sets of context-trials in which the retinal direction of the stimulus is matched between intra-saccadic and post-saccadic context conditions. Alternatively, the context trials could be kept the way they were done in Experiment 2, but now the test trials in both intra-saccadic and post-saccadic context conditions could have one of two retinal directions: the direction imposed on the retina during saccades (vertical + horizontal) or the direction imposed during fixation (purely vertical). These experiments would reveal if the habituation is indeed specific to saccades or if the specificity to saccades found in the first two experiments is merely a consequence of differences in retinal motion trajectories. Moreover, the experiment should be run with a balanced design; i.e., habituation to upwards and downwards motion should be compared.

I thank the reviewer for giving me the chance to provide a clear answer on that question. I have carried out as the reviewers suggested. I describe the details of this experiment in the corresponding Materials and methods and Results sections.

[Editors' note: further revisions were suggested prior to acceptance, as described below.]

The reviewers appreciate that you have added new experiments to address the concerns raised in the previous round. They think that the findings from Experiment 3 are convincing, but that the findings from the Supplementary experiment are problematic for your proposal that this study deals with motion discrimination rather than displacement discrimination. Based on these discussions and the individual reviews below, we are inviting you to resubmit your manuscript, we are proposing a way forward for you to consider.

I agree with the reviewers that the data from the supplementary experiment alone do not provide sufficient evidence to decide whether this study deals with motion suppression. I now discuss this explicitly in the manuscript. Furthermore, I have made changes in the Abstract and Discussion accordingly. I have restructured the Discussion section and added a paragraph stating that the current data do not specify which visual features trigger habituation. I mention saturation of motion-sensitive neurons as one possible mechanism followed by a paragraph outlining the idea of displacement detection via pre-and post saccadic image comparisons. I now write:

“However, from the present data it cannot be determined how stimulus-specific transsaccadic habituation is. […] The experimental task that I applied can in principle be solved without motion information.”

and:

“In the experiments of the present study, it cannot be decided to what extent participants employed a comparison between the pre-and post-saccadic image and how much they used the motion transient to judge the displacement.”

Experiment 3 seems convincing because it tries to match saccade-induced and screen motion, uses a balanced design (both upwards and downwards context motions are tested), and provides converging evidence for the directional tuning of the contextual habituation effect. Thus, the results of Experiment 2 are unlikely to be a consequence of differences in retinal motion vectors between different context conditions.

I agree and stick to the initial statements regarding the directional tuning.

In the supplementary experiment a full-field random pattern mask flashed during saccade execution diminishes performance (comparison of pre- and post-saccadic target locations). The author concludes that because the mask abolished the motion transient, it is the motion transient that drives performance.However, there is a serious logical flaw in this experiment. The results seem to be inconsistent with the author's theoretical proposal; i.e., that saccadic omission results from habituation to the motion transients that accompany every saccade (and therefore there is no need for special trans-saccadic mechanisms as claimed by many in the literature). In support of this claim, Experiment 1 shows that exposing observers to artificial intra-saccadic motion degrades motion discrimination performance (relative to performance without exposure to artificial intra-saccadic motion). The supplementary experiment is essentially the same as Experiment 1, except that now the upwards motion is masked by a full-field random pattern (although it still remains to be shown whether such a pattern actually does mask motion during normal viewing, i.e. fixation). If habituation to intra-saccadic motion explains poor motion discrimination, then elimination of intra-saccadic motion should release discrimination performance. Performance should therefore improve or, at least, remain comparable to the no-exposure condition (because there is, of course, still the usual saccade-induced motion transient in the no-exposure condition). Instead, results show the reverse – performance gets worse – which cannot easily be reconciled with the theoretical claim the author is making. Even if we consider that the mask itself generates motion on the retina during the saccade, the vector of that motion is horizontal and should not – according to the author's proposal – affect vertical motion detection.

Please note that the mask was only displayed in test trials. Please see my description in the previous version of the manuscript:

“The presentation of the grating, fixation point and saccade target in the context trials was identical to Experiment 1. However, in contrast to Experiment 1 the mask presentation and the grating displacement in the test trials was triggered by saccade detection and occurred 35 ms after saccade onset in every trial (see Figure 3—figure supplement 1A,B).”

If the mask would have been presented in the context trials, then indeed, there would be the logical flaw that the reviewers pointed out. The idea for this experiment was to find out how well subjects can detect displacements if the motion transients is covered by the mask. The reviewers suggest that the efficiency of the mask has to be demonstrated in a fixation experiment. I agree. When setting up the experiment, I saw that – even in fixation – discriminating the displacement was very hard. I took this to be line with earlier work by Sabine Born and me. But surely, this should be tested for the exact stimulus that I used in the present saccade experiment.

It is therefore doubtful how much the supplementary experiment adds with regard to the main finding of the current manuscript (the plasticity). I am convinced the logic of the experiment can be defended, but again more experiments would be necessary. I therefore have decided to remove the supplementary experiment.

Indeed, in the rebuttal, the author argues that his results do speak to motion perception because detecting the displacement change across saccades requires visibility of the motion transient. In support of his argument, he cites Shiori and Cavanagh, 1989, who showed that sensitivity to the motion transient was abolished during saccades. But this doesn't make much sense: if the motion transient is not perceived during the saccade, how can it be used to drive displacement reports?

I think the reason for the confusion my statements induced might be the usage of terms like “absence of motion transients” or “insensitivity to displacements”. Saccadic suppression is never a complete insensitivity to contrast or motion transients. In my own data presented in this manuscript, psychometric functions could be estimated, clearly suggesting that subjects did perceive the displacement to some degree.

He does not cite (but could have) experiments performed by Deubel and Schneider in which they showed that performance increases dramatically when there is a blank (that diminishes the motion transient) between pre- and post-saccadic targets. So it seems that there is evidence that diminishing the motion transient may actually help trans-saccadic location performance.

I agree that this is a classic study that should be cited and I thank the reviewers for pointing this out. However, I disagree with the reviewer’s interpretation of this study. In the Deubel and Schneider’s “no blank" condition, the motion transient fell into the saccadic period and was thereby diminished. Consequently, subjects showed poor discrimination performance. When there was a blank, the target reappeared after the saccadic period, thus making the motion transient visible again. Under this condition, subjects saw the displacements well.

Previous work from the author and Sabine Born showed that it is possible to mimic poor performance in a fixation task in which a mask occurs between the two targets. However, being able to degrade performance in two different ways (either by presenting targets before and after a saccade, or by presenting them before and after a mask) does not mean that the same processes are involved.

The argument of our articles on that topic was that by using the mask we degraded performance in a way similar enough to trans-saccadic stimulation to conclude that the same process is involved. It is true that a full field mask is not the best match for what is on the retina during the saccade but it might be close enough. This result shows that the loss of the motion transient decreases discrimination of displacements. But I see the reviewer’s point, one could mimic the saccadic disturbance better with a mirror that moves with saccadic speed. This condition could also be used to test the alternative interpretation that the reviewers mention in the following point below. As I have decided to remove the supplementary experiment, this experiment has to be carried out in future work.

An alternative interpretation of the results is that the intra-saccadic mask might induce uncertainty about the initial position of the grating, making the displacement harder to judge (rather than rendering the motion invisible); for instance, the masking pattern may have itself induced transients on the retina that caused the system to down-regulate any input during subsequent saccades. This would argue that the author's protocol does not study motion perception (as expressed in the title of the manuscript), but displacement detection.

This alternative interpretation might be compatible with the model by Crevecoeur and Körding, 2017 which states that suppression scales with the unreliability of trans-saccadic vision. As I have dropped the supplementary experiment, this issue is not discussed further in the new version of the manuscript.

To justify the conclusion that is the motion transient that drives performance, two further pieces of information would be needed. (1) Does such a full-field mask abolish a motion transient of an embedded target (in a fixation condition)?

I agree that such a control is necessary. Please also see my reply above.

2) Why is the invisibility of the motion transient during a saccade (as shown by Shiori and Cavanagh) not sufficient?

In most tests saccadic suppression is not complete. We are not functionally blind during saccades as often claimed. For motion stimuli, it has been shown that the trans-saccadic visibility depends on the stimulus characteristics (Castet and Masson, 2000). The same holds true for saccadic suppression of contrast: point-like stimuli are detected less than background changes (Brook and Fuchs, 1975). In the study of Shiori and Cavanagh performance was at chance level. However, their stimulus (random dot motion) was much different than my grating displacement. As a matter of fact, there was no complete suppression of the grating displacement in the current study, otherwise I would not have been able to estimate a psychometric function.

The author would have to provide evidence that the manipulation does indeed reduce motion perception while leaving the ability to judge displacements intact.

Indeed, the ability to judge displacements by comparing pre-and post-saccadic images should be left intact after reductions of motion perception. Performance could be compared to trials in which isoluminant stimuli are used as they are not “seen” by the motion detectors. I will pursue this issue in a follow-up study.

In sum, the only condition in which performance increased after context trials was the context in which the author minimized changes on the retina during the saccade (no displacement context trials). It seems, therefore, that any visual changes during saccades result in insensitivity during subsequent saccades. The plasticity in intrasaccadic omission remains an interesting finding, but it deviates quite strongly from the claim in the title/Abstract/conclusion that this paper is about saccade suppression of motion. Thus, we invite you to provide an alternative interpretation to saccadic suppression of motion for your current findings. Note that this would entail a re-writing of the entire manuscript, theory included. Please note that in eLife it is not customary to have many review rounds; thus, if you decide to take this route please try to make a comprehensive and clear a case as possible.

I have followed this advice and removed all instances of “motion suppression” from the title, Abstract and the main text. As the supplementary experiment is judged inconclusive the statement “that any visual changes during saccades result in insensitivity” is not really shown by the present data. I keep the term “saccadic suppression” instead of “saccadic omission”

AnalysisIn several of the new results, the author conducted multiple t-tests rather than an ANOVA, which is not good statistical practice. Take as an example: in Experiment 2, the author tests for direction-specificity of the contextual habituation with four conditions resulting from the crossing of two factors, Direction of Motion in Test Trials (Upward/Downward) and Time of shift (intra-sacc: 35 ms/post-sacc: 98 ms). In the results he only presents t-test on the Direction Factor, independently for each time of shift. Because the t-test is significant in intra-sacc, and not in post-sacc, the author concludes that there is direction-specific intra-saccadic habituation. Of course, this conclusion only holds if there is a statistically reliable interaction between the two factors. Factorial designs should be analyzed with ANOVAs throughout, instead of t-tests. To respond to the sample size critique, the author could use a linear mixed model with p-values estimated by the methods explained by Luke (2017, Behav Research Methods), which apparently is less sensitive to sample size than ANOVAs and t-tests.

Many thanks for pointing this article out! I have now calculated non-parametric ANOVAs and estimated the p-values with the Kenward-Roger approximations. I describe the analysis in the section on data analysis:

“For statistical analysis of Experiments 2 and 3, a non-parametric repeated measures ANOVA was calculated, using the Aligned Rank Transform (Wobbrock et al., 2011). Significance was determined by applying the Kenward-Roger approximation to estimate p-values, a procedure that has been shown to produce acceptable Type 1 error rates even for smaller samples (Luke, 2017).”

One issue that further complicates the interpretation of the results of the supplementary experiment is that it is difficult to have an idea of power. How big of a difference in slope can 4 subjects reveal (given the variability seen in the other experiments, and especially when several cannot be fit by a cumulative Gaussian)?

I agree. For this reason and the other argument listed above, I decided to drop that experiment from the manuscript.